# Different clinical impact of hyperuricemia according to etiologies of chronic kidney disease: Gonryo Study

**Kimio Watanabe**[1]*, **Masaaki Nakayama**[1,2], **Tae Yamamoto**[3], **Gen Yamada**[4], **Hiroshi Sato**[5], **Mariko Miyazaki**[6,7], **Sadayoshi Ito**[7,8]

1 Division of Kidney Center, St Luke's International Hospital, Tokyo, Japan, 2 Research Division of Chronic Kidney Disease and Dialysis Treatment, Tohoku University Hospital, Sendai, Japan, 3 Division of Kidney Center, Sendai City Hospital, Sendai, Japan, 4 Division of Nephrology and Endocrinology, Osaki Citizen Hospital, Osaki, Japan, 5 Division of Internal Medicine, JR Sendai Hospital, Sendai, Japan, 6 Division of Blood Purification, Tohoku University Hospital, Sendai, Japan, 7 Division of Nephrology, Hypertension and Endocrinology, Tohoku University, Sendai, Japan, 8 Katta General Public Hospital, Shiroishi, Japan

* spyh63x9@gmail.com

## Abstract

### Background

Hyperuricemia is highly prevalent in chronic kidney disease (CKD) patients, but the evidence for a relationship between uric acid (UA) and clinical outcomes in CKD patients is limited and inconsistent. We hypothesized that UA has a different impact on clinical outcomes according to the underlying disease causing CKD.

### Methods

This study prospectively investigated the associations between UA and renal and non-renal outcomes according to the underlying disease causing CKD in 2,797 Japanese patients under the care of nephrologists. The patients were categorized into four groups: primary renal disease (n = 1306), hypertensive nephropathy (n = 467), diabetic nephropathy (n = 275), and other nephropathy (n = 749). The renal outcome was defined as end-stage renal disease (ESRD), and the non-renal outcome was defined as a composite endpoint of cardiovascular events (CVEs) and all-cause mortality.

### Results

During a median 4.8-year follow-up, 359 (12.8%) patients reached the renal outcome, and 260 (9.3%) reached the non-renal outcome. In the all-patient analysis, hyperuricemia was not associated with the risks for renal and non-renal outcomes, but in primary renal disease (PRD) and hypertensive renal disease (HTN) patients, hyperuricemia was significantly associated with non-renal outcomes. Per 1 mg/dl higher UA level, multivariable adjusted hazard ratio was 1.248 (95% CI: 1.003 to 1.553) for PRD, and 1.250 (1.035 to 1.510) for HTN. Allopurinol did not reduce the risks for renal and non-renal outcomes, both in all patients and in the subgroup analysis.

**Data Availability Statement:** All relevant data are within the manuscript and its Supporting Information files.

**Funding:** The authors received no specific funding for this work.

**Competing interests:** The authors have declared that no competing interests exist.

## Conclusions

The effect of hyperuricemia on clinical outcomes in CKD patients varies according to the underlying disease causing CKD. Hyperuricemia is an independent risk factor for non-renal outcomes in primary renal disease and hypertensive renal disease patients. Allopurinol did not decrease the risks for renal and non-renal outcomes.

## Introduction

Previous epidemiological studies of the general population showed the independent effect of hyperuricemia on the risk for renal outcomes, such as development of chronic kidney disease (CKD), or non-renal outcomes, such as cardiovascular events (CVEs) and deaths [1–6]. However, the evidence regarding the relationship between uric acid (UA) and clinical outcomes in CKD patients is limited and inconsistent [7–11].

Other than UA, multiple and specific risk factors including decreased GFR, urinary protein, renal anemia, and CKD-related mineral bone disorder also exist in CKD patients, and differences in the underlying diseases causing CKD, such as glomerulonephritis, hypertensive renal disease, and diabetic nephropathy, might affect clinical outcomes [12–14]. Several studies performed in subpopulations have shown the effect of hyperuricemia on the risk for progression of renal dysfunction [15–17], but how different are the impacts of UA on risks for renal and non-renal outcomes by the underlying disease causing CKD is unknown.

Furthermore, the amount of data on whether allopurinol, a urate-lowering agent, improves renal and non-renal outcomes in CKD is limited and inconsistent [18–23].

Based on this background, the aim of this study was to clarify the effects of hyperuricemia and allopurinol on the risk of renal outcomes, specifically end-stage renal disease (ESRD), and non-renal outcomes, specifically the composite endpoint of CVEs and all-cause mortality, according to the difference in the underlying disease causing CKD in a large, multicenter, prospective, observational cohort study (Gonryo Study).

## Methods

### Study protocol and ethics statement

The Gonryo CKD cohort is a prospective, observational survey to clarify long-term clinical outcomes of outpatients followed by nephrologists in 11 hospitals, including Tohoku University Hospital in Miyagi Prefecture, which is located in the north-east area of Japan. Between May 2006 and November 2008, 4015 patients who had given their written, informed consent were included and followed until December 2015. The study, which was performed in accordance with the principles of the Helsinki Declaration, was approved by the institutional review board (IRB) of Tohoku University School of Medicine and the respective participating hospitals (Number 2006–10, UMIN000011211). Nephrologists and their assistants in participating hospitals kept a record of each participants' outcomes once a year in medical records and picked up them for analysis. The National Kidney Foundation Kidney Disease Outcomes Quality Initiative Guidelines were applied to define CKD, and CKD stage was classified by patients' baseline estimated GFR (eGFR), calculated using the Modification of Diet in Renal Disease (MDRD) study equation for Japanese people: eGFR (mL/min/1.73 m$^2$) = 194 x serum creatinine (-1.094) x age (-0.287) x 0.739 (if female) [24]. Subjects were excluded from the study for the following reasons: no-show (n = 1),

eGFR > 60 mL/min/1.73 m$^2$ without proteinuria (n = 837), eGFR > 60 mL/min/1.73 m$^2$ without urine test (n = 94), and no data for serum UA (n = 125). Patients who have continued to use each drug at the time of entry are registered as medication users, and, inclusion criteria as drug user are not set by the length of the administration period. Finally, 2,797 patients meeting the CKD criteria were evaluated in the present study (Fig 1). Hyperuricemia was defined the serum UA level of 7.0 mg/dL or higher regardless of sex. To identify the pathogenetic importance of hyperuricemia for clinical outcomes, all subjects were classified according to the underlying cause of CKD into four categories: primary renal disease (PRD, n = 1,306; biopsy proven in 78.6%); hypertensive nephropathy (HTN, n = 467; biopsy proven in 12.8%); diabetic nephropathy (DN, n = 275; biopsy proven in 17.1%); and other nephropathy (Others, n = 749; biopsy proven in 37.8%). Hypertensive nephropathy was defined by preceding history of hypertension with absence of other possible disorders, including cases with biopsy findings of nephrosclerosis, and diabetic nephropathy was defined by preceding history of diabetes accompanying nephropathy with absence of other possible renal disorders, including cases with biopsy findings of diabetic nephropathy or those who presenting nephropathy with diabetic retinopathy in the absence of other

## Enrolled, N = 4,015

### Excluded

837 eGFR > 60 mL/min/1.73 m$^2$ without proteinuria

94 eGFR > 60 mL/min/1.73 m$^2$ without urine test

1 No-show

## Met CKD criteria, N = 2,922

### Excluded

125 Without serum uric acid data

## Inclusion, N = 2,797

**Fig 1. Patient selection in the present study.**

possible disorders. Other nephropathies included vasculitis, polycystic kidney disease, genetic disorder, interstitial nephritis, and CKD of unknown cause. The baseline characteristics of all patients and each group are shown in Table 1.

**Table 1. Baseline characteristics of 2,797 CKD patients, overall and by underlying disease causing CKD.**

| Characteristic | | Underlying disease causing CKD | | | |
|---|---|---|---|---|---|
| | Overall | PRD | HTN | DN | Others |
| N | 2797 | 1306 | 467 | 275 | 749 |
| Age (y) | 60.5±16.2 | 56.0±16.8 | 70.0±11.6 | 66.6±12.7 | 59.9±15.6 |
| Male | 1521 (54.4) | 723 (55.4) | 269 (57.6) | 180 (65.5) | 348 (46.6) |
| CKD stage | | | | | |
| Stage 1+2 | 1050 (37.5) | 625 (47.9) | 77 (16.5) | 53 (19.3) | 295 (39.4) |
| Stage 3a | 602 (21.5) | 265 (20.3) | 172 (36.8) | 40 (14.5) | 125 (16.7) |
| Stage 3b | 453 (16.2) | 196 (15.0) | 86 (18.4) | 39 (14.2) | 132 (17.6) |
| Stage 4 | 403 (14.4) | 137 (10.5) | 80 (17.1) | 59 (21.5) | 127 (16.9) |
| Stage 5 | 285 (10.2) | 80 (6.1) | 52 (11.1) | 84 (30.5) | 69 (9.2) |
| CVD and CKD risk factors | | | | | |
| Hypertension | 2186 (78.2) | 965 (73.9) | 467 (100) | 249 (90.5) | 531 (70.9) |
| Diabetes | 766 (27.4) | 204 (15.6) | 149 (31.9) | 275 (100) | 138 (18.4) |
| Cardiac disease | 359 (12.8) | 106 (8.1) | 104 (22.3) | 66 (24.0) | 83 (11.1) |
| Stroke | 180 (6.4) | 46 (3.5) | 56 (11.9) | 34 (12.4) | 44 (5.9) |
| Dyslipidemia | 1202 (43.0) | 579 (44.3) | 195 (41.8) | 144 (52.4) | 284 (37.9) |
| Smoking | 455 (16.3) | 210 (16.1) | 75 (16.1) | 57 (20.7) | 113 (15.1) |
| Creatinine (mg/dL) | 1.5±1.4 | 1.3±1.2 | 1.6±1.3 | 2.5±1.9 | 1.5±1.2 |
| eGFR (mL/min/1.73 m$^2$) | 53.3±29.7 | 59.9±29.6 | 44.7±22.6 | 35.9±26.3 | 53.5±31.0 |
| Urinary protein* | 1398 (50.0) | 642 (49.2) | 234 (50.1) | 215 (78.2) | 307 (40.9) |
| Hemoglobin (g/dL) | 12.7±2.1 | 13.1±1.9 | 12.6±2.2 | 11.5±2.3 | 12.5±2.1 |
| Albumin (g/dL) | 4.0±0.5 | 4.1±0.5 | 4.1±0.4 | 3.7±0.6 | 3.9±0.5 |
| LDL (mg/dL) | 111.5±31.9 | 110.1±30.9 | 110.6±30.1 | 111.5±33.6 | 114.5±33.9 |
| SBP (mmHg) | 131.4±16.2 | 129.5±15.2 | 134.9±17.4 | 136.9±17.7 | 130.4±15.9 |
| DBP (mmHg) | 76.7±10.8 | 77.2±10.3 | 76.2±11.6 | 73.9±11.7 | 77.1±10.7 |
| PP (mmHg) | 54.7±12.7 | 52.3±11.2 | 58.7±13.7 | 62.9±14.6 | 53.3±11.9 |
| Medication use | | | | | |
| ACE inhibitors or ARBs | 1778 (63.6) | 819 (62.7) | 338 (72.4) | 212 (77.1) | 409 (54.6) |
| Diuretics | 447 (15.9) | 125 (9.6) | 90 (19.3) | 129 (46.9) | 103 (13.8) |
| Allopurinol | 691 (24.7) | 290 (22.2) | 148 (31.7) | 64 (23.3) | 189 (25.2) |
| Statins | 963 (34.4) | 467 (35.8) | 148 (31.7) | 119 (43.3) | 229 (30.6) |
| Steroid | 682 (24.4) | 419 (32.1) | 13 (2.8) | 6 (2.2) | 224 (29.9) |
| Antiplatelets | 1265 (45.2) | 725 (55.5) | 160 (34.3) | 115 (41.8) | 265 (35.4) |
| Uric acid | | | | | |
| Total (mg/dL) | 6.4±1.7 | 6.3±1.6 | 6.5±1.8 | 7.0±2.1 | 6.2±1.7 |
| Male (mg/dL) | 6.8±1.6 | 6.7±1.4 | 6.9±1.7 | 7.2±2.0 | 6.7±1.6 |
| Female (mg/dL) | 5.8±1.8 | 5.7±1.6 | 5.9±1.8 | 6.7±2.2 | 5.7±1.7 |
| Renal biopsy | 1416 (50.6) | 1026 (78.6) | 60 (12.8) | 47 (17.1) | 283 (37.8) |

PRD, primary renal disease; HTN, hypertensive renal disease; DN, diabetic nephropathy; CKD, chronic kidney disease; CVD, cardiovascular disease; eGFR, estimated glomerular filtration; LDL, low-density lipoprotein; SBP, systolic blood pressure; DBP, diastolic blood pressure; PP, pulse pressure; ACE, angiotensin-converting enzyme; ARB, angiotensin II receptor blocker.

*Urinary protein means the number and proportion of patients who were positive on the dipstick urinary protein test.

## Outcomes and measurements

The study outcomes included ESRD (initiation of hemodialysis or peritoneal dialysis), CVEs, and all-cause mortality. CVEs included angina pectoris, acute myocardial infarction, congestive heart failure, and stroke (cerebral bleeding or infarction). Outcomes were determined from medical records, death certificates, and interviews with attending physicians at the time of the patients' annual checkups until the fifth-year follow-up. In cases with congestive heart failure, only those who needed admission for treatment were counted. Asymptomatic cerebral infarction or lacunar infarction was not included. Baseline data obtained from medical records at each hospital at the beginning of the study included demographic data, laboratory data, smoking habits, body mass index (BMI), underlying disease causing CKD with information about renal biopsy, comorbid condition (hypertension, diabetes mellitus, dyslipidemia, coronary artery disease, cerebrovascular disease, peripheral artery disease), blood pressure, and heart rate. Laboratory data included blood creatinine, hemoglobin, hematocrit, blood urea nitrogen, UA, albumin, calcium, phosphate, cholesterol, triglycerides, C-reactive protein, and urinary protein by dipstick test for spot urine. Drug use information included angiotensin-converting enzyme (ACE) inhibitors, angiotensin II receptor blockers (ARBs), calcium channel blockers, beta blockers, other antihypertensive agents, loop diuretics, thiazide diuretics, aldosterone antagonists, antidiabetic drugs including insulin, lipid-lowering agents, anti-platelet agents, corticosteroids, activated vitamin D, phosphate binders, erythropoiesis stimulating agents, intravenous or oral iron use, and allopurinol. These data were collected annually from cohort entry until the end of follow-up.

## Statistical analysis

Statistical analyses were performed using SPSS version 22.0 (IBM, Tokyo, Japan). Data are shown as means ± standard deviation (SD) for continuous variables, and categorical variables are shown as numbers and percentage. A $p$ value less than 0.05 was considered significant. The Chi-squared test or Fisher's exact test was used for differences in categorical variables, and analysis of variance (ANOVA) or the Kruskal-Wallis test was used for continuous data. We analyzed UA as a continuous variable and investigated associations between UA per 1 mg/dl higher value with ESRD, CVEs, and all-cause mortality. A Cox proportional hazard model was used to evaluate the relationships between UA and ESRD, CVEs, and all-cause mortality. Analyzed variables included underlying disease of CKD, eGFR, urinary protein, systolic blood pressure, past history of cardiac disease, ACE inhibitor and/or ARB use, diuretic use, statins, antiplatelets, age, sex, smoking habits, BMI, hemoglobin, and albumin. Data were censored at the time of death, initiation of hemodialysis or peritoneal dialysis, or the end of the study in December 2015. Survival analysis was performed using the Kaplan-Meier method with the stratified log-rank test.

## Results

### Baseline characteristics and the association between UA level and CKD stage

The baseline characteristics of the patients in the present study are summarized in Table 1. A total of 2,797 patients were analyzed, with a mean follow-up of 4.8 years. The mean age of all subjects was 60.5±16.2 years, with men accounting for 54.4%. Mean eGFR was 53.3±29.7 mL/min/1.73 m$^2$, UA was 6.4±1.7 mg/dL, and 24.7% of patients were on allopurinol at the start of the study. In terms of underlying diseases causing CKD, age was higher with HTN, whereas eGFR was lower and baseline uric acid was higher in DN. The UA level was higher in men

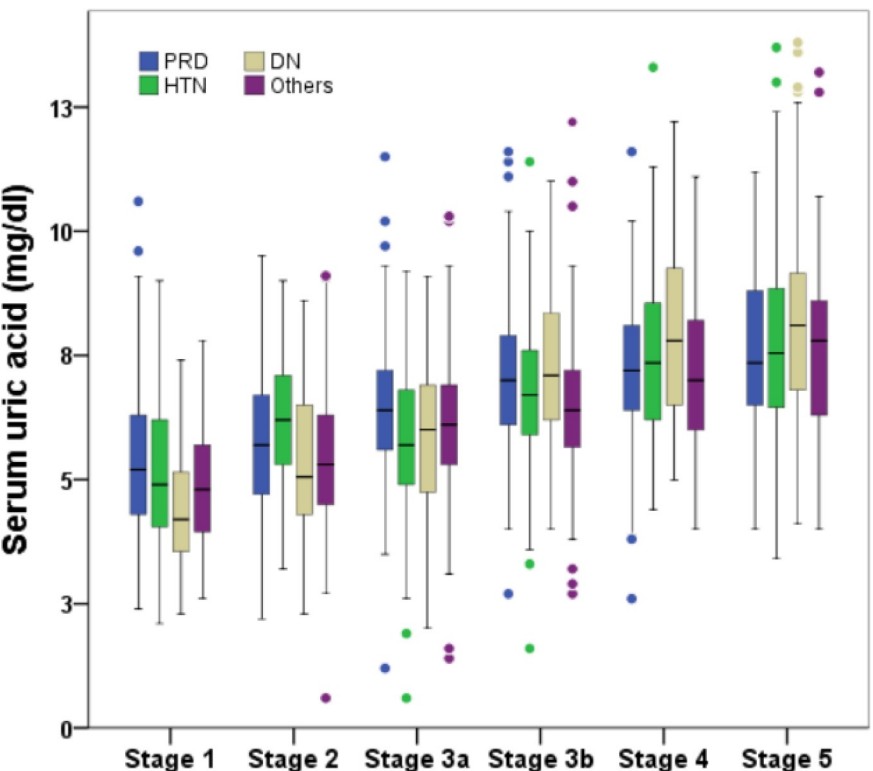

**Fig 2. Boxplot of baseline serum uric acid levels by CKD stage according to underlying disease causing CKD.**
PRD, primary renal disease; HTN, hypertensive renal disease; DN, diabetic nephropathy.

than in women regardless of the cause of CKD. An increasing trend in both the mean UA level and the proportion of patients with a UA level greater than 8.0 mg/dL was confirmed with increasing CKD stage (Fig 2).

## Uric acid and renal outcome (ESRD)

A total of 359 patients (12.8%) reached the renal outcome (ESRD). In all subjects, UA was not an independent risk factor for ESRD with the Cox proportional hazard model adjusted for underlying disease causing CKD, eGFR, urinary protein, systolic blood pressure, past history of cardiac disease, ACE inhibitors/ARBs, diuretics, statins, antiplatelets, age, sex, smoking, BMI, hemoglobin, and albumin. In the subgroup analysis, only sex (male) was associated with an increased risk for ESRD (adjusted HR per 1 mg/dl higher UA level was 1.101, [95%CI 1.009–1.202], p = 0.031) (Fig 3A).

## Uric acid and non-renal outcomes (CVEs and deaths)

There were 260 patients (9.3%) who reached the non-renal outcome (composite endpoint of CVEs and all-cause mortality). In all subjects, UA was not an independent risk factor for the non-renal outcome with the Cox proportional hazard model adjusted for underlying disease causing CKD, eGFR, urinary protein, systolic blood pressure, past history of cardiac disease, ACE inhibitors/ARBs, diuretics, statins, antiplatelets, age, sex, smoking, BMI, hemoglobin, and albumin. In the subgroup analysis, UA was associated with an increased risk for the non-renal outcome in PRD and HTN group. Per 1 mg/dl higher UA level, multivariable adjusted

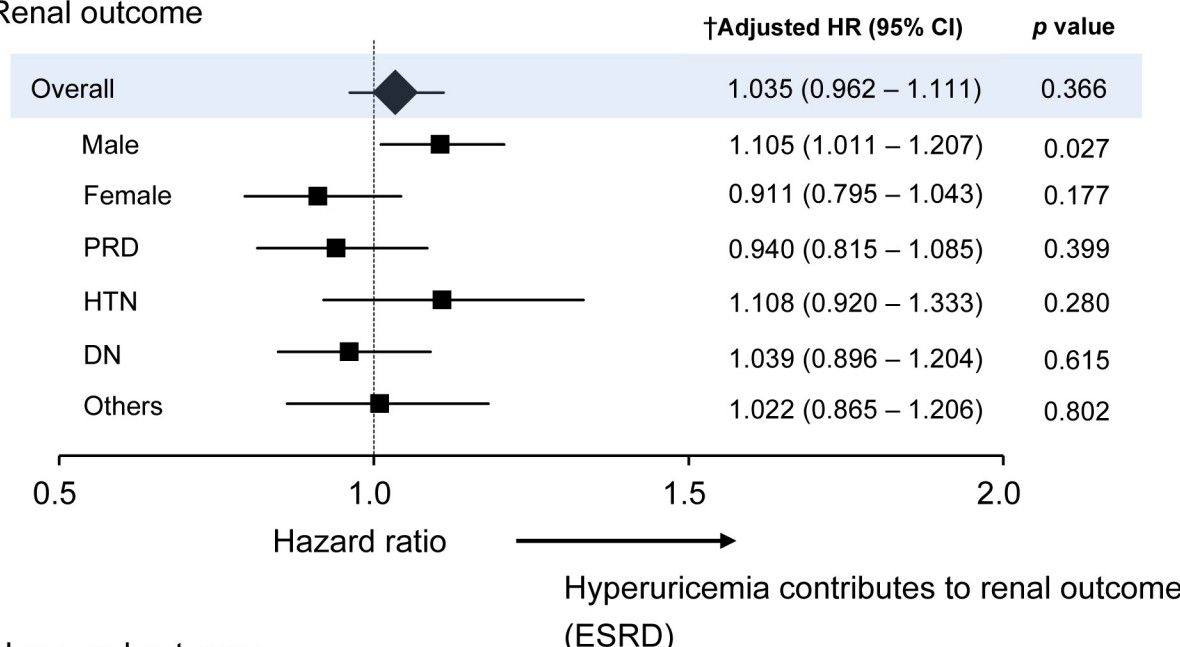

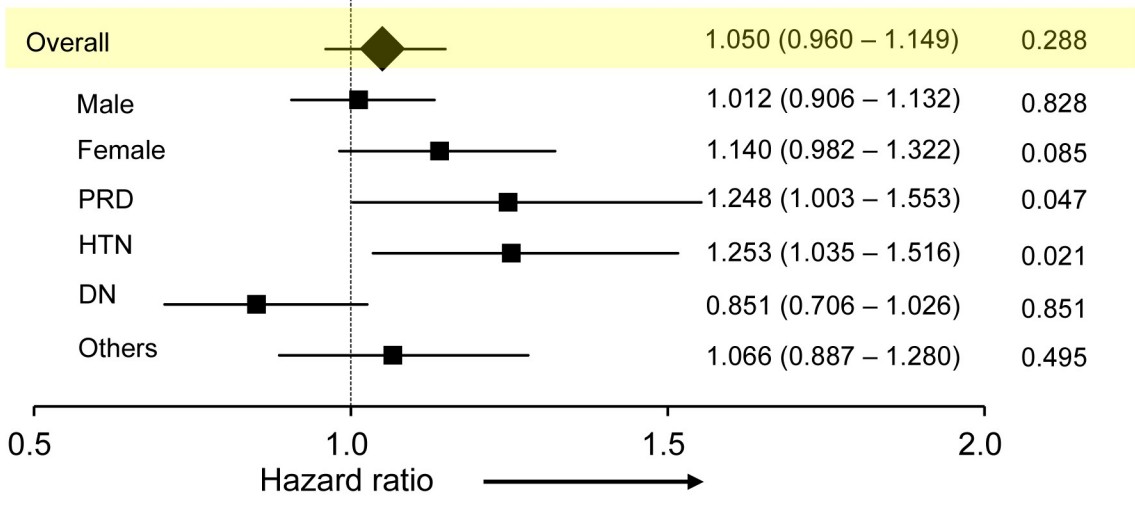

**Fig 3. Associations of uric acid with renal and non-renal outcomes.** CI, confidence interval; HR, hazard ratio; PRD, primary renal disease; HTN, hypertensive renal disease; DN, diabetic nephropathy; CVEs, cardiovascular events. †Associations are measured per 1 mg/dl increment of UA using Cox proportional hazard regression. Hazard ratios (HRs) were adjusted for underlying disease of CKD, eGFR, urinary protein, systolic blood pressure, past history of cardiac disease, ACEi/ARBs, diuretics, statins, antiplatelets, allopurinol, age, sex, smoking, body mass index, hemoglobin, albumin.

HR was 1.248 (95% CI: 1.003 to 1.553) for PRD, and 1.250 (1.035 to 1.510) for HTN (**Fig 3B**). UA 6.0 mg/dL in PRD and UA 6.8 mg/dL in HTN are optimal cut off value from ROC analysis. AUC of ROC curve and p-value are 0.611 (95%CI, 0.537–0.685, p = 0.004) in PRD and 0.658 (95%CI, 0.584–0.732, p = 0.012) in HTN. Detailed data are also shown in **S1 Table**.

### Effect of allopurinol on renal and non-renal outcomes

A total of 697 (24.7%) patients were prescribed allopurinol at baseline. The efficacy of allopurinol for preventing renal and non-renal outcomes was not confirmed in both all patients and in the subgroup analysis (Fig 4 and S2 Table). Kaplan-Meier analysis with the stratified log-rank test was performed to determine the efficacy of allopurinol for preventing renal and non-renal outcomes (Fig 5).

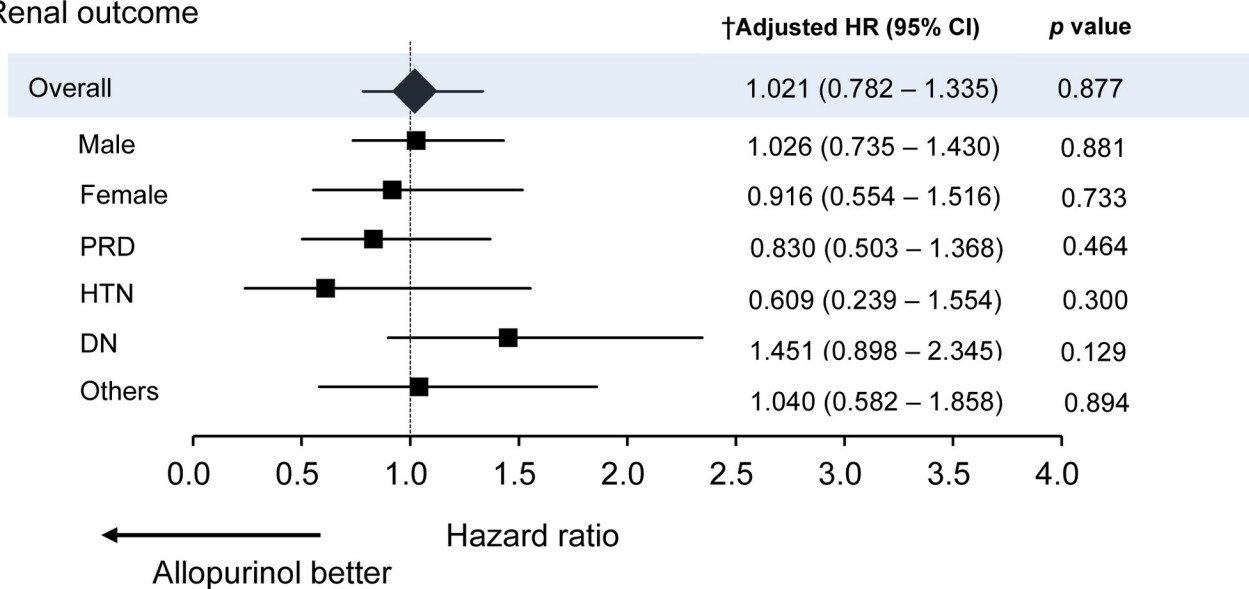

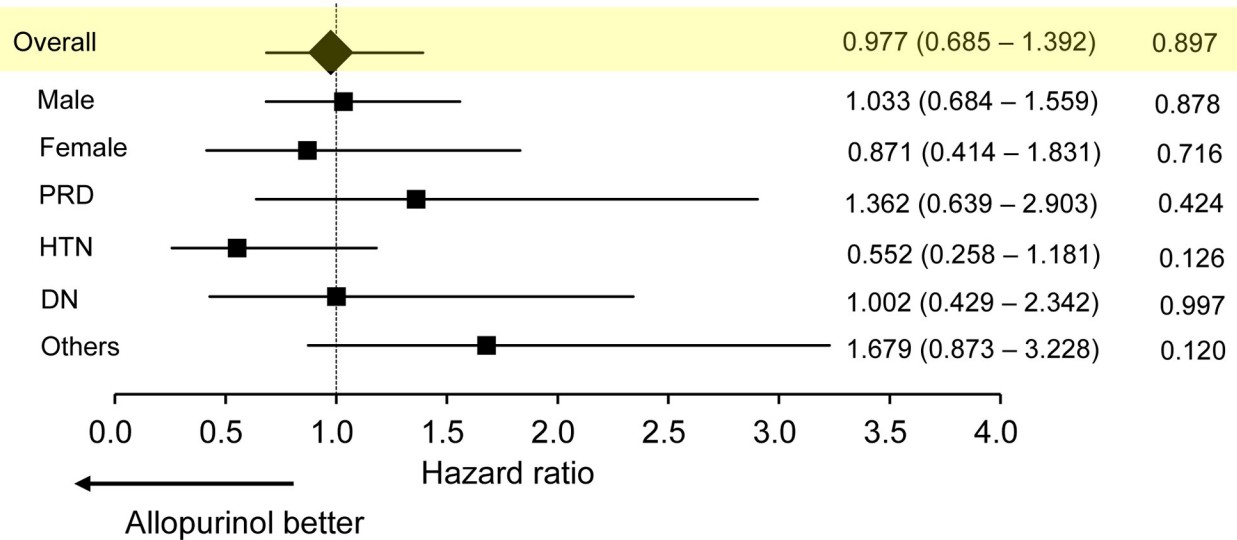

**Fig 4. Associations of allopurinol with renal and non-renal outcomes.** CI, confidence interval; HR, hazard ratio; PRD, primary renal disease; HTN, hypertensive renal disease; DN, diabetic nephropathy; CVEs, cardio-vascular events. †Associations are measured per 1 mg/dl increment of UA using Cox proportional hazard regression. Hazard ratios (HRs) were adjusted for underlying disease of CKD, eGFR, urinary protein, systolic blood pressure, past history of cardiac disease, ACEi/ARBs, diuretics, statins, antiplatelets, allopurinol, age, sex, smoking, body mass index, hemoglobin, albumin.

## Discussion

The results of this study demonstrated that hyperuricemia is an independent risk factor for non-renal outcomes in primary renal disease and hypertensive renal disease patients, and allopurinol did not decrease the risk in these patients. We analyzed UA as a continuous variable and demonstrated that per 1 mg/dl higher UA level, multivariable adjusted HR was 1.248 (95% CI: 1.003 to 1.553) for PRD, and 1.250 (1.035 to 1.510) for HTN. The strength of the study is that it clarified the long-term prognosis of CKD outpatients followed by nephrologists according to the underlying disease causing CKD in a large-scale, multi-center, prospective cohort. We previously showed the beneficial effect of allopurinol in 178 HTN patients with impaired kidney function (eGFR< 45 mL/min/1.73 m$^2$) during 18.4-month follow-up [25], but this result was not confirmed in the present study. The reason for this discrepancy is that a larger number of variables was included in the present analysis than in the previous study. The number of patients who reached non-renal outcomes in the present study and the previous one was 359 and 28, respectively. The more accurate analysis with adjustment of more variables in the present study showed the efficacy of allopurinol to be unclear.

The potential role of UA in clinical outcomes has been debated for decades [11], and hyperuricemia is thought to have an impact, not large, but certainly related to adverse events in CKD patients based on various previous studies [9, 10, 12–14, 17, 26]. Of course, hyperuricemia is a commonly seen finding in CKD, and it could be a consequence of reduced excretion or diuretic agents [27]. Based on evidence concerning UA and possible links to hypertension, renal disease, and cardiovascular disease [5], hyperuricemia in hypertensive renal disease appears to be an important issue, given the present results. Basic research can help us understand the specific mechanisms of this relationship: higher UA levels cause endothelial dysfunction, activation of the renin-angiotensin aldosterone system, vascular injury, tubulointerstitial inflammation, and elevated blood pressure [9, 28]. In the present study, 63.6% of the patients were taking ACE inhibitors or ARBs at baseline, which could have attenuated the potential effects of hyperuricemia on the activity of the renin-angiotensin system.

IgA nephropathy is the most common primary glomerulonephritis in Japan [29]. Although it has been suggested that IgA nephropathy patients with hyperuricemia have poor renal prognosis [12, 17], the relationship between hyperuricemia in IgA nephropathy patients and non-renal outcome has been unclear. We confirmed that non-renal outcome in PRD is poor in this study, however, the data of detailed classification of PRD is lacking. It is necessary to clarify how the renal and non-renal outcomes differs depending on the type of PRD in the future.

Several animal studies have demonstrated that allopurinol treatment prevented the development of hypertension, structural and functional alterations in the glomerular afferent arteriole, and the ischemic type of renal parenchymal injury via modulation of the renin-angiotensin system and neuronal NO synthase [30, 31]. However, recent multicenter, randomized, double-blind, placebo-controlled trials (RCTs), such as "Febuxostat Versus Placebo Randomized Controlled Trial Regarding Reduced Renal Function in Patients With Hyperuricemia Complicated by Chronic Kidney Disease Stage 3" (FEATHER), the Preventing Early Renal Loss in Diabetes (PERL), and "the Controlled Trial of Slowing of Kidney Disease Progression from the Inhibition of Xanthine Oxidase" (CKD-FIX) failed to demonstrate clinically meaningful benefits of serum urate reduction with allopurinol or febuxostat on kidney function decline, similar to the present results [21, 22, 32]. In addition, the KDIGO (Kidney Disease: Improving Global Outcomes) guideline clinical practice recommendation says that, "there is insufficient evidence to support or refute the use of agents to lower serum uric acid concentrations in people with CKD and either symptomatic or asymptomatic hyperuricemia in order to delay progression of CKD" [33]. On the other hand, hyperuricemia has been

**(a) Renal outcome (ESRD)/All patients**

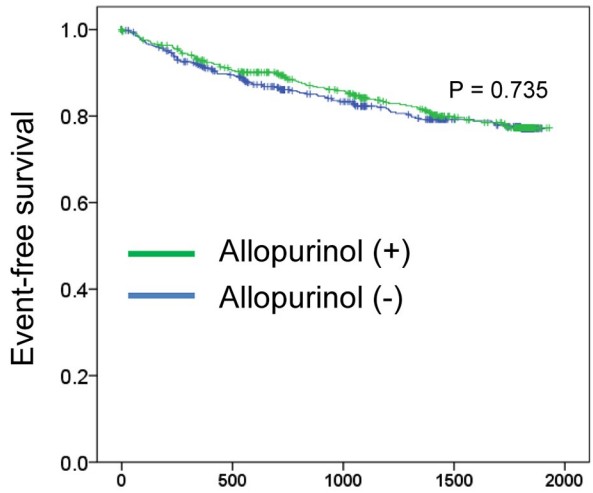

**(b) Non-renal outcome (CVEs and deaths)/all patients**

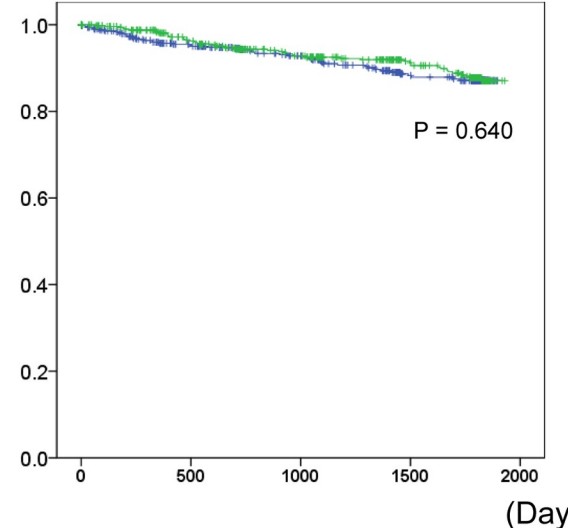

**(c) Renal outcome (ESRD)/HTN patients**

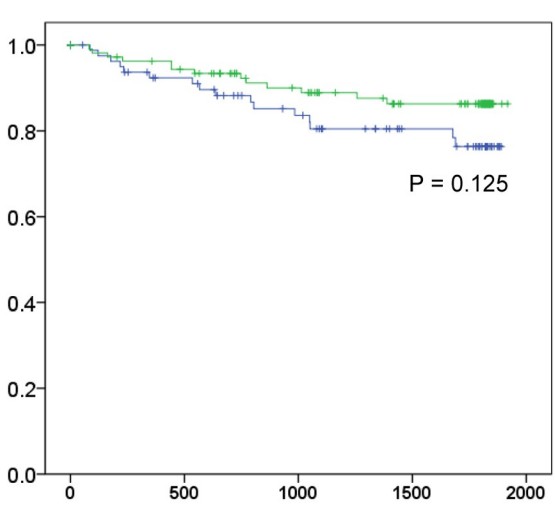

**(d) Non-renal outcome (CVEs and deaths)/HTN patients**

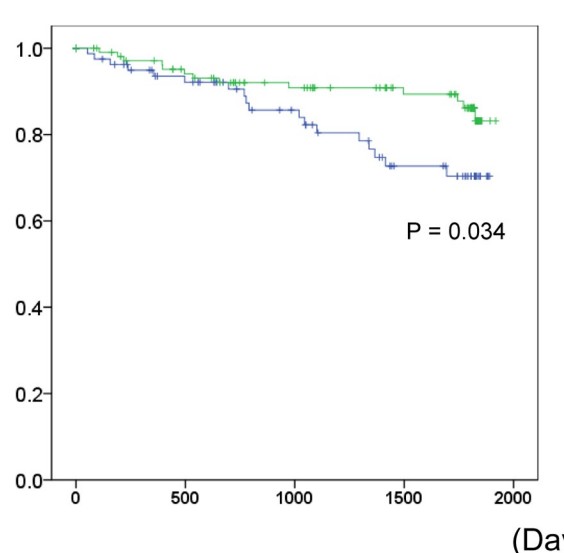

**Fig 5. Impact of allopurinol on renal and non-renal outcomes.**

shown to be a risk factor for cardiovascular disease, including myocardial infarction and stroke [34]. Based on various findings so far, including the present study, we should carefully consider whether we should treat asymptomatic hyperuricemia and the relevance of lowering uric acid levels. In addition, further studies are needed to clarify which CKD patients, e.g. underlying disease causing CKD, severity of CKD stage, and degree of uric acid elevation, would benefit from treatment.

The present study has several limitations. First, since this was an observational study, it only shows an association between uric acid or allopurinol and clinical outcomes, not a cause and effect relationship. Second, details of the adjustment of allopurinol, such as

discontinuation, reduction, and increase, were not available also, the data of switching to other XOR inhibitors, such as febuxostat or topiroxostat in the treatment course is lacking. Third, renal biopsy was performed only 12.8% in HTN group, and 17.1% in DN group, and nearly one-third patients in the HTN group had diabetes mellitus. It is considered difficult to distinguish between nephrosclerosis and DKD in a strict sense without biopsy. So, caution should be taken in interpreting the data.

## Conclusions

The effect of hyperuricemia on clinical outcomes in CKD patients varies according to the underlying disease causing CKD. Hyperuricemia is an independent risk factor for non-renal outcomes in primary renal disease and hypertensive renal disease patients, and allopurinol did not decrease the risks for renal and non-renal outcomes.

## Supporting information

**S1 Table. Associations of uric acid with renal and non-renal outcomes.**
(DOCX)

**S2 Table. Associations of allopurinol with renal and non-renal outcomes.**
(DOCX)

## Author Contributions

**Conceptualization:** Kimio Watanabe, Masaaki Nakayama, Mariko Miyazaki.

**Data curation:** Kimio Watanabe.

**Formal analysis:** Kimio Watanabe, Mariko Miyazaki.

**Investigation:** Kimio Watanabe, Mariko Miyazaki.

**Methodology:** Masaaki Nakayama, Tae Yamamoto, Mariko Miyazaki.

**Supervision:** Masaaki Nakayama, Tae Yamamoto, Gen Yamada, Hiroshi Sato, Mariko Miyazaki, Sadayoshi Ito.

**Writing – original draft:** Kimio Watanabe.

**Writing – review & editing:** Masaaki Nakayama, Tae Yamamoto, Gen Yamada, Hiroshi Sato, Mariko Miyazaki, Sadayoshi Ito.

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
