## [Decision Letter · Decision Letter 0]

18 Jan 2021

PONE-D-20-34099

Relationships of underlying disease causing chronic kidney disease to hyperuricemia and clinical outcomes: Gonryo study

PLOS ONE

Dear Dr. Kimio Watanabe,

Thank you for submitting your manuscript to PLOS ONE. After careful consideration, we feel that it has merit but does not fully meet PLOS ONE’s publication criteria as it currently stands. Therefore, we invite you to submit a revised version of the manuscript that addresses the points raised during the review process.

We look forward to receiving your revised manuscript.

Kind regards,

Ping-Hsun Wu, M.D. PhD.

Academic Editor

PLOS ONE

Additional Editor Comments:

The method to link follow-up outcomes should be described. Some important factors are missing in the adjusted model, such as Statin. Please add the potential confounders in the model or acknowledge them as a limitation.

Journal Requirements:

2. In your Methods section, please provide additional information about the participant recruitment method and the demographic details of your participants. Please ensure you have provided sufficient details to replicate the analyses such as: a) a description of how participants were recruited, and b) descriptions of how samples were collected.

3. Please provide a sample size and power calculation in the Methods, or discuss the reasons for not performing one before study initiation.

Reviewers' comments:

Reviewer's Responses to Questions

**Comments to the Author**

1. Is the manuscript technically sound, and do the data support the conclusions?

Reviewer #1: Yes

Reviewer #2: Partly

2. Has the statistical analysis been performed appropriately and rigorously? 

Reviewer #1: Yes

Reviewer #2: Yes

3. Have the authors made all data underlying the findings in their manuscript fully available?

Reviewer #1: Yes

Reviewer #2: No

4. Is the manuscript presented in an intelligible fashion and written in standard English?

Reviewer #1: Yes

Reviewer #2: Yes

5. Review Comments to the Author

Reviewer #1: Watanabe et al. investigated the effect of hyperuricemia on clinical outcomes in different CKD etiology. Hyperuricemia is an independent risk factor for non-renal outcomes in hypertensive renal disease. I have some comments to improve the study.

1. Primary renal disease was the major cause of CKD in the Gonryo study (2006 -2008). I wondered the proportion of major cause of CKD in Japan.

2. The present in both figures and tables is redundant in the main text. Please move the detailed data of Table to the supplementary material.

3. Page 15, line 203, the full name of “RCT”, “FEATHER”, “PERL”, and “CKD-FIX” should be presented.

4. Since hyperuricemia is associated with non-renal outcomes in hypertensive renal disease patients, a spline function for uric acid level and non-renal outcomes will be interesting to observe the adequate and optimal uric acid cut off value.

5. The observation subjects were enrolled between May 2006 and November 2008 and follow up until December 2015. How to link these participants and recorded their outcomes? Is there any national registry or database to link subjects’ outcomes?

Reviewer #2: The authors demonstrated the association between hyperuricemia and clinical outcomes in CKD patients in a prospective observational cohort study. It has novelty because the authors showed the effects of hyperuricemia varies according to underlying diseases causing CKD. However, there are some issues needed to be clarified.

Major issue:

Q1: The authors should explain in more detail how to classify CKD patients into different categories, especially when patients had more than two underlying diseases causing CKD simultaneously. According to table 1, 149(31.9%) patients in hypertensive nephropathy group(HTN) had diabetes mellitus, and 249(90.5%) patients in diabetic nephropathy group(DN) had hypertension. However, only 12.8% and 17.1% patients in HTN and DN respectively received renal biopsy. Without biopsy, how did the authors judge and confirm the underlying diseases causing CKD?

Q2: Statins have been proven to be effective in the primary and secondary prevention of cardiovascular disease. However, statins were not adjusted in the evaluation of non-renal outcomes (CVEs and deaths) in the present study. Meanwhile, information about the use of antiplatelet agents was not available. The lack of these important factors may cause biases in evaluation of CVE.

Minor issue:

Q3: What is the definition of hyperuricemia? What is the cut-point value of hyperuricemia used in the current study? Was the hyperuricemia defined according to the initial single laboratory data or the average of serial blood exams during follow-up? Were patients divided into quartiles according to serum UA level? The authors should define hyperuricemia more clearly.

Q4: How to define medication use? Did a single prescription of medication fit the definition of medication use? Or a continuous use of a drug for a period of time could be regarded as medication use. The authors should define medication use more clearly.

Q5: In the current study, allopurinol did not decrease the risks for renal and non-renal outcomes. However, information about the use of other UA lowering agents, such as febuxostat and benzbromarone, was not available. That may lead to biases when evaluating the effects of UA lowering agent and clinical outcomes. Is it possible to consider all the UA lowering agents in the current study?

Q6: The title of the present study “ Relationships of underlying disease causing chronic kidney disease to hyperuricemia and clinical outcomes “ was a little bit confusing. Could the authors make it more clear?

6. PLOS authors have the option to publish the peer review history of their article (what does this mean?). If published, this will include your full peer review and any attached files.

Reviewer #1: No

Reviewer #2: No

---

## [Author Response · Author response to Decision Letter 0]

8 Feb 2021

Additional Editor Comments:

The method to link follow-up outcomes should be described. Some important factors are missing in the adjusted model, such as Statin. Please add the potential confounders in the model or acknowledge them as a limitation.

[Response]

National registry or database to link subjects’ outcomes do not exist in Japan. Nephrologists and their assistants in eleven participating hospitals in Miyagi Prefecture kept a record of each participants’ outcomes once a year in medical records and picked up them for analysis. Regarding this point, we added the following comment in Methods section.

Line 80-81, Nephrologists and their assistants in participating hospitals kept a record of each participants’ outcomes once a year in medical records and picked up them for analysis.

Following your comment, we re-analyzed the adjusted model adding statins and antiplatelets as the potential confounders. Please, see figures and tables (Fig. 3 and 4, S1 and S2 Table). As the result of re-analysis, we identified primary renal disease (PRD) patients also have the risk for non-renal outcomes. Therefore, we modified manuscript as follows. 

Line 40-43, but in primary renal disease (PRD) and hypertensive renal disease (HTN) patients, hyperuricemia was significantly associated with non-renal outcomes on multivariate Cox proportional analysis (adjusted hazard ratio [95% confidence interval] PRD, 1.248 [1.003 – 1.553]; HTN, 1.250 [1.035 – 1.510]).

Line 47, Hyperuricemia is an independent risk factor for non-renal outcomes in primary renal disease and hypertensive renal disease patients.

Line 172-173, In the subgroup analysis, only PRD and HTN was were associated with an increased risk for the non-renal outcome (HR 1.250, 95%CI 1.035-1.510, p=0.020 adjusted HR [95% Cl] PRD, 1.248 [1.003 – 1.553]; HTN, 1.250 [1.035 – 1.510]) (Figure 3-b).

Line 187-189, The results of this study demonstrated that hyperuricemia is an independent risk factor for non-renal outcomes in primary renal disease and hypertensive renal disease patients, and allopurinol did not decrease the risk in these patients.

Line 247-249, Hyperuricemia is an independent risk factor for non-renal outcomes in primary renal disease and hypertensive renal disease patients, and allopurinol did not decrease the risks for renal and non-renal outcomes.

Line 211-216, IgA nephropathy is the most common primary glomerulonephritis in Japan [29]. Although it has been suggested that IgA nephropathy patients with hyperuricemia have poor renal prognosis [12, 17], the relationship between hyperuricemia in IgA nephropathy patients and non-renal outcome was unclear. We confirmed that non-renal outcome in PRD is poor in this study, however, the data of detailed classification of PRD is lacking. It is necessary to clarify how the renal and non-renal outcomes differs depending on the type of PRD in the future. 

[29] Sugiyama H, Yokoyama H, Sato H, Saito T, Kohda Y, Nishi S, et al. Japan Renal Biopsy Registry and Japan Kidney Disease Registry: Committee Report for 2009 and 2010. Clin Exp Nephrol 2013; 17(2): 155-73

Reviewer #1: Watanabe et al. investigated the effect of hyperuricemia on clinical outcomes in different CKD etiology. Hyperuricemia is an independent risk factor for non-renal outcomes in hypertensive renal disease. I have some comments to improve the study.

1. Primary renal disease was the major cause of CKD in the Gonryo study (2006 -2008). I wondered the proportion of major cause of CKD in Japan.

[Response to 1]

Unfortunately, there are no Japanese patients’ data showing the proportion of major cause of CKD. Since the Gonryo study targets only CKD patients who had been followed by nephrologist, the proportion of nephritis patients is thought to be relatively high compared to the actual situation. The most common primary cause of renal failure among new dialysis patients was diabetic nephropathy (43.8%), followed by chronic glomerulonephritis (32.4%) in the nationwide survey conducted at the end of 2013 in Japan*. Based on the result, it is expected that there would actually be more DN/DKD patients compared to our result in Table1.

*Ikuto M, Shigeru N, Satoshi O, Naoki K, Norio H, Takayuki H, et al. An Overview of Regular Dialysis Treatment in Japan (As of 31 December 2013). Ther Apher Dial 2015; 19(6): 540-74

2. The present in both figures and tables is redundant in the main text. Please move the detailed data of Table to the supplementary material.

[Response to 2]

We moved the detailed data of Table (Table 2 and 3) to the supplementary material (S1 and S2 Table). 

3. Page 15, line 203, the full name of “RCT”, “FEATHER”, “PERL”, and “CKD-FIX” should be presented.

[Response to 3]

We corrected the sentence which you pointed out as follows. 

Line 221-227, However, recent RCTs, such as FEATHER, PERL, and CKD-FIX, However, recent multicenter, randomized, double-blind, placebo-controlled trials (RCTs), such as “Febuxostat Versus Placebo Randomized Controlled Trial Regarding Reduced Renal Function in Patients With Hyperuricemia Complicated by Chronic Kidney Disease Stage 3” (FEATHER), the Preventing Early Renal Loss in Diabetes (PERL), and “the Controlled Trial of Slowing of Kidney Disease Progression from the Inhibition of Xanthine Oxidase” (CKD-FIX) failed to demonstrate clinically meaningful benefits of serum urate reduction with allopurinol or febuxostat on kidney function decline, similar to the present results [21][22][32].

4. Since hyperuricemia is associated with non-renal outcomes in hypertensive renal disease patients, a spline function for uric acid level and non-renal outcomes will be interesting to observe the adequate and optimal uric acid cut off value.

[Response to 4]

We detected that UA 6.0 mg/dL in PRD and UA 6.8 mg/dL in HTN are optimal cut off value from ROC analysis. Regarding this point, we added the following comment in Result section.

Line 174-176, UA 6.0 mg/dL in PRD and UA 6.8 mg/dL in HTN are optimal cut off value from ROC analysis. AUC of ROC curve and p-value are 0.611 (95%CI, 0.537-0.685, p=0.004) in PRD and 0.658 (95%CI, 0.584-0.732, p=0.012) in HTN. 

5. The observation subjects were enrolled between May 2006 and November 2008 and follow up until December 2015. How to link these participants and recorded their outcomes? Is there any national registry or database to link subjects’ outcomes?

 [Response to 5]

National registry or database to link subjects’ outcomes do not exist in Japan. Nephrologists and their assistants in eleven participating hospitals in Miyagi Prefecture kept a record of each participants’ outcomes once a year in medical records and picked up them for analysis. Regarding this point, we added the following comment in Methods section.

Line 80-81, Nephrologists and their assistants in participating hospitals kept a record of each participants’ outcomes once a year in medical records and picked up them for analysis.

Reviewer #2: The authors demonstrated the association between hyperuricemia and clinical outcomes in CKD patients in a prospective observational cohort study. It has novelty because the authors showed the effects of hyperuricemia varies according to underlying diseases causing CKD. However, there are some issues needed to be clarified.

Major issue:Q1: The authors should explain in more detail how to classify CKD patients into different categories, especially when patients had more than two underlying diseases causing CKD simultaneously. According to table 1, 149(31.9%) patients in hypertensive nephropathy group(HTN) had diabetes mellitus, and 249(90.5%) patients in diabetic nephropathy group(DN) had hypertension. However, only 12.8% and 17.1% patients in HTN and DN respectively received renal biopsy. Without biopsy, how did the authors judge and confirm the underlying diseases causing CKD?

[Response to Q1]

Patients were classified according to one of four underlying renal diseases diagnosed by the attending physicians at the participating hospitals. Hypertensive nephropathy was defined by preceding history of hypertension with absence of other possible disorders, including cases with biopsy findings of nephrosclerosis, and diabetic nephropathy was defined by preceding history of diabetes accompanying nephropathy with absence of other possible renal disorders, including cases with biopsy findings of diabetic nephropathy or those who presenting nephropathy with diabetic retinopathy in the absence of other possible disorders. Following your suggestion, we added the detailed explanation in the methods section.

Line 95-100, Hypertensive nephropathy was defined by preceding history of hypertension with absence of other possible disorders, including cases with biopsy findings of nephrosclerosis, and diabetic nephropathy was defined by preceding history of diabetes accompanying nephropathy with absence of other possible renal disorders, including cases with biopsy findings of diabetic nephropathy or those who presenting nephropathy with diabetic retinopathy in the absence of other possible disorders.

Q2: Statins have been proven to be effective in the primary and secondary prevention of cardiovascular disease. However, statins were not adjusted in the evaluation of non-renal outcomes (CVEs and deaths) in the present study. Meanwhile, information about the use of antiplatelet agents was not available. The lack of these important factors may cause biases in evaluation of CVE.

[Response to Q2]

Following your comment, we re-analyzed the adjusted model adding statins and antiplatelets as the potential confounders. As the result of re-analysis, we identified primary renal disease (PRD) patients also have the risk for non-renal outcomes. Please, see figures and tables (Fig. 3 and 4, S1 and S2 Table). Also we modified the manuscript. 

Minor issue:Q3: What is the definition of hyperuricemia? What is the cut-point value of hyperuricemia used in the current study? Was the hyperuricemia defined according to the initial single laboratory data or the average of serial blood exams during follow-up? Were patients divided into quartiles according to serum UA level? The authors should define hyperuricemia more clearly.

[Response to Q3]

First of all, hyperuricemia in this report was defined the serum uric acid level of 7.0 mg/dL or higher regardless of sex. Regarding this point, we added the following comment in Methods section.

Line 91, Hyperuricemia was defined the serum UA level of 7.0 mg/dL or higher regardless of sex.

And we demonstrate 1-mg/dL increase in uric acid level is associated with a hazard ratio of 1.250 [1.035 – 1.510] for non-renal outcome in hypertensive renal disease patients. In this analysis, we used uric acid level at the time of enrollment as continuous variable. And we detected that UA 6.0 mg/dL in PRD and UA 6.8 mg/dL in HTN are optimal cut off value from ROC analysis. Regarding this point, we added the following comment in Result section.

Line 174-176, UA 6.0 mg/dL in PRD and UA 6.8 mg/dL in HTN are optimal cut off value from ROC analysis. AUC of ROC curve and p-value are 0.611 (95%CI, 0.537-0.685, p=0.004) in PRD and 0.658 (95%CI, 0.584-0.732, p=0.012) in HTN.

Q4: How to define medication use? Did a single prescription of medication fit the definition of medication use? Or a continuous use of a drug for a period of time could be regarded as medication use. The authors should define medication use more clearly. 

[Response to Q4]

Patients who have continued to use allopurinol at the time of entry were registered as medication users. Therefore, inclusion criteria as drug user are not set by the length of the administration period. Regarding this point, we added the following comment in Methods section.

Line 88-90, Patients who have continued to use each drug at the time of entry are registered as medication users, and inclusion criteria as drug user are not set by the length of the administration period.

Q5: In the current study, allopurinol did not decrease the risks for renal and non-renal outcomes. However, information about the use of other UA lowering agents, such as febuxostat and benzbromarone, was not available. That may lead to biases when evaluating the effects of UA lowering agent and clinical outcomes. Is it possible to consider all the UA lowering agents in the current study?

[Response to Q5]

In the current study, UA level and allopurinol users are extracted and analyzed only at the time of entry (from 2006 to 2008). And very few patients were using uric acid lowering agents other than allopurinol. For this reason, febuxostat, which has been available since 2011 in Japan, or benzbromarone are not considered in this analysis. Of course, it is expected that there are not a few cases in which allopurinol was switched to febuxostat or topiroxostat during the course of treatment, and possibility of bias is considered, so this point was added to the limitation. 

Line 240-242, Second, details of the adjustment of allopurinol, such as discontinuation, reduction, and increase, were not available, also, the data of switching to other XOR inhibitors, such as febuxostat or topiroxostat in the treatment course is lacking.

Q6: The title of the present study “ Relationships of underlying disease causing chronic kidney disease to hyperuricemia and clinical outcomes “ was a little bit confusing. Could the authors make it more clear?

[Response to Q6]

We changed the title of the present study from “Relationships of underlying disease causing chronic kidney disease to hyperuricemia and clinical outcomes” to “Different clinical impact of hyperuricemia according to etiologies of chronic kidney disease: Gonryo Study”.

---

## [Decision Letter · Decision Letter 1]

5 Mar 2021

PONE-D-20-34099R1

Different clinical impact of hyperuricemia according to etiologies of chronic kidney disease: Gonryo Study

PLOS ONE

Dear Dr. Kimio Watanabe,

Thank you for submitting your manuscript to PLOS ONE. After careful consideration, we feel that it has merit but does not fully meet PLOS ONE’s publication criteria as it currently stands. Therefore, we invite you to submit a revised version of the manuscript that addresses the points raised during the review process.

We look forward to receiving your revised manuscript.

Kind regards,

Ping-Hsun Wu, M.D. PhD.

Academic Editor

PLOS ONE

Journal Requirements:

Additional Editor Comments (if provided):

The clinical classification of CKD etiology could be described clearly. A suggestion for the revised method and result section for adding statin and the anti-platelet variable was considered.

Reviewers' comments:

Reviewer's Responses to Questions

**Comments to the Author**

1. If the authors have adequately addressed your comments raised in a previous round of review and you feel that this manuscript is now acceptable for publication, you may indicate that here to bypass the “Comments to the Author” section, enter your conflict of interest statement in the “Confidential to Editor” section, and submit your "Accept" recommendation.

Reviewer #1: All comments have been addressed

Reviewer #2: (No Response)

2. Is the manuscript technically sound, and do the data support the conclusions?

Reviewer #1: Yes

Reviewer #2: Yes

3. Has the statistical analysis been performed appropriately and rigorously? 

Reviewer #1: Yes

Reviewer #2: Yes

4. Have the authors made all data underlying the findings in their manuscript fully available?

Reviewer #1: Yes

Reviewer #2: No

5. Is the manuscript presented in an intelligible fashion and written in standard English?

Reviewer #1: Yes

Reviewer #2: Yes

6. Review Comments to the Author

Reviewer #1: (No Response)

Reviewer #2: Q1: The authors added the definition of HTN and DN in the revised manuscript. However, the authors did not mention about how to classify if a patient had both diabetes mellitus and hypertension simultaneously. As shown in Table 1, nearly one-third patients in the HTN group had diabetes mellitus. Why were these patients regarded as HTN rather than DN? Since low biopsy rate in patients from HTN and DN groups, more convincing and detailed classification criteria to differentiate HTN from DN should be described. Otherwise, the authors should mention that misclassification is possible and the data should be interpreted with caution in the limitation section.

Q2: Since the authors analyzed UA as a continuous variable, and the relationship between UA and clinical outcomes was demonstrated as hazard ratios per 1 mg/dl greater UA level, they should mention this clearly in the Methods section, and described the results more precisely in the manuscript (Abstract, Results section, Discuss section, and caption of figures)

Q3: The authors re-analyzed the HR after taking statin and anti-platelet into consideration in the revised manuscript. In the Statistical analysis sections and Results section, adjusting for statin and anti-platelet should be mentioned. Please make sure that all corresponding description in the manuscript have been modified.

Q4: In line 172-173, The sentence "In the subgroup analysis, PRD and HTN was associated with an increased risk for the non-renal outcome" was confusing. Please make it more clear.

7. PLOS authors have the option to publish the peer review history of their article (what does this mean?). If published, this will include your full peer review and any attached files.

Reviewer #1: No

Reviewer #2: No

---

## [Author Response · Author response to Decision Letter 1]

8 Mar 2021

Reviewer #1: (No Response)

Reviewer #2: Q1: The authors added the definition of HTN and DN in the revised manuscript. However, the authors did not mention about how to classify if a patient had both diabetes mellitus and hypertension simultaneously. As shown in Table 1, nearly one-third patients in the HTN group had diabetes mellitus. Why were these patients regarded as HTN rather than DN? Since low biopsy rate in patients from HTN and DN groups, more convincing and detailed classification criteria to differentiate HTN from DN should be described. Otherwise, the authors should mention that misclassification is possible and the data should be interpreted with caution in the limitation section.

[Response]

Patients who have not had renal biopsy and who have both hypertension and diabetes at the same time were classified as either HTN or DN by the attending physicians’ judgment based on the medical history, urine test and presence or absence of diabetic retinopathy. Hypertensive nephropathy was defined by preceding history of hypertension with absence of other possible disorders, including cases with biopsy findings of nephrosclerosis, and diabetic nephropathy was defined by preceding history of diabetes accompanying nephropathy with absence of other possible renal disorders, including cases with biopsy findings of diabetic nephropathy or those who presenting nephropathy with diabetic retinopathy in the absence of other possible disorders. However, biopsy rate is low in HTN and DN. And it is considered difficult to distinguish between HTN and DN in a strict sense without biopsy finding. We think this point is considered as the limitation of this study, so we added the following comments. 

Line 247-250 (Discussion) Third, renal biopsy was performed in 50.6% of all patients, and only 12.8% of hypertensive renal disease patients. Third, renal biopsy was performed only 12.8% in HTN group, and 17.1% in DN group, and nearly one-third patients in the HTN group had diabetes mellitus. It is considered difficult to distinguish between HTN and DN in a strict sense without biopsy. So, caution should be taken in interpreting the data. 

Q2: Since the authors analyzed UA as a continuous variable, and the relationship between UA and clinical outcomes was demonstrated as hazard ratios per 1 mg/dl greater UA level, they should mention this clearly in the Methods section, and described the results more precisely in the manuscript (Abstract, Results section, Discuss section, and caption of figures)

[Response]

We modified the manuscript about description for relationship between UA and clinical outcomes as follows. 

Line 42-43, (Abstract) on multivariate Cox proportional analysis (adjusted hazard ratio [95% confidence interval] PRD, 1.248 [1.003 – 1.553]; HTN, 1.250 [1.035 – 1.510]). Per 1 mg/dl higher UA level, multivariable adjusted hazard ratio was 1.248 (95% CI: 1.003 to 1.553) for PRD, and 1.250 (1.035 to 1.510) for HTN. 

Line 139-141, (Methods) We analyzed UA as a continuous variable and investigated associations between UA per 1 mg/dl higher value with ESRD, CVEs, and all-cause mortality.

Line 166-167 (Results) risk for ESRD (adjusted HR per 1 mg/dl higher UA level was 1.101, [95%CI 1.009-1.202], p=0.031)

Line 174-177 (Results) In the subgroup analysis, UA was associated with an increased risk for the non-renal outcome in PRD and HTN group. PRD and HTN was associated with an increased risk for the non-renal outcome (adjusted HR [95% Cl] PRD, 1.248 [1.003 – 1.553]; HTN, 1.250 [1.035 – 1.510]) Per 1 mg/dl higher UA level, multivariable adjusted HR was 1.248 (95% CI: 1.003 to 1.553) for PRD, and 1.250 (1.035 to 1.510) for HTN.

Line 192-194 (Discussion) We analyzed UA as a continuous variable and demonstrated that per 1 mg/dl higher UA level, multivariable adjusted HR was 1.248 (95% CI: 1.003 to 1.553) for PRD, and 1.250 (1.035 to 1.510) for HTN. 

Legends for Figure 3, Adjusted for underlying disease of CKD, eGFR, urinary protein, systolic blood pressure, past history of cardiac disease, ACEis/ARBs, diuretics, statins, antiplatelets, allopurinol, age, sex, smoking, body mass index, hemoglobin, albumin. Associations are measured per 1 mg/dl increment of UA using Cox proportional hazard regression. Hazard ratios (HRs) were adjusted for underlying disease of CKD, eGFR, urinary protein, systolic blood pressure, past history of cardiac disease, ACEi/ARBs, diuretics, statins, antiplatelets, allopurinol, age, sex, smoking, body mass index, hemoglobin, albumin.

Legends for Figure 4, Adjusted for underlying disease of CKD, eGFR, urinary protein, systolic blood pressure, past history of cardiac disease, ACEis/ARBs, diuretics, statins, antiplatelets, allopurinol, age, sex, smoking, body mass index, hemoglobin, albumin. Associations are measured per 1 mg/dl increment of UA using Cox proportional hazard regression. Hazard ratios (HRs) were adjusted for underlying disease of CKD, eGFR, urinary protein, systolic blood pressure, past history of cardiac disease, ACEi/ARBs, diuretics, statins, antiplatelets, allopurinol, age, sex, smoking, body mass index, hemoglobin, albumin.

Q3: The authors re-analyzed the HR after taking statin and anti-platelet into consideration in the revised manuscript. In the Statistical analysis sections and Results section, adjusting for statin and anti-platelet should be mentioned. Please make sure that all corresponding description in the manuscript have been modified.

[Response]

We checked all corresponding description in the manuscript and modified as follows. 

Line 142-145 (Methods) Analyzed variables included underlying disease of CKD, eGFR, urinary protein, systolic blood pressure, past history of cardiac disease, ACE inhibitor and/or ARB use, diuretic use, statins, antiplatelets, age, sex, smoking habits, BMI, hemoglobin, and albumin.

Line 162-166 (Results) In all subjects, UA was not an independent risk factor for ESRD with the Cox proportional hazard model adjusted for underlying disease causing CKD, eGFR, urinary protein, systolic blood pressure, past history of cardiac disease, ACE inhibitors/ARBs, diuretics, statins, antiplatelets, age, sex, smoking, BMI, hemoglobin, and albumin.

Line 171-174 (Results) In all subjects, UA was not an independent risk factor for the non-renal outcome with the Cox proportional hazard model adjusted for underlying disease causing CKD, eGFR, urinary protein, systolic blood pressure, past history of cardiac disease, ACE inhibitors/ARBs, diuretics, statins, antiplatelets, age, sex, smoking, BMI, hemoglobin, and albumin.

Q4: In line 172-173, The sentence "In the subgroup analysis, PRD and HTN was associated with an increased risk for the non-renal outcome" was confusing. Please make it more clear.

[Response]

We modified the sentence as follows according to the suggestion. 

Line 174-176, In the subgroup analysis, PRD and HTN was associated with an increased risk for the non-renal outcome UA was associated with an increased risk for the non-renal outcome in PRD and HTN group

---

## [Decision Letter · Decision Letter 2]

15 Mar 2021

Different clinical impact of hyperuricemia according to etiologies of chronic kidney disease: Gonryo Study

PONE-D-20-34099R2

Dear Dr. Kimio Watanabe,

We’re pleased to inform you that your manuscript has been judged scientifically suitable for publication and will be formally accepted for publication once it meets all outstanding technical requirements.

Kind regards,

Ping-Hsun Wu, M.D. PhD.

Academic Editor

PLOS ONE

Additional Editor Comments (optional):

All suggestions had been revised accordingly. This manuscript is available for publication.

Reviewers' comments:

Reviewer's Responses to Questions

**Comments to the Author**

1. If the authors have adequately addressed your comments raised in a previous round of review and you feel that this manuscript is now acceptable for publication, you may indicate that here to bypass the “Comments to the Author” section, enter your conflict of interest statement in the “Confidential to Editor” section, and submit your "Accept" recommendation.

Reviewer #2: All comments have been addressed

2. Is the manuscript technically sound, and do the data support the conclusions?

Reviewer #2: Yes

3. Has the statistical analysis been performed appropriately and rigorously? 

Reviewer #2: Yes

4. Have the authors made all data underlying the findings in their manuscript fully available?

Reviewer #2: No

5. Is the manuscript presented in an intelligible fashion and written in standard English?

Reviewer #2: Yes

6. Review Comments to the Author

Reviewer #2: (No Response)

7. PLOS authors have the option to publish the peer review history of their article (what does this mean?). If published, this will include your full peer review and any attached files.

Reviewer #2: No

---

## [Editor Report · Acceptance letter]

17 Mar 2021

PONE-D-20-34099R2 

Different clinical impact of hyperuricemia according to etiologies of chronic kidney disease: Gonryo Study 

Dear Dr. Watanabe:

I'm pleased to inform you that your manuscript has been deemed suitable for publication in PLOS ONE. Congratulations! Your manuscript is now with our production department. 

Kind regards, 

on behalf of

Dr. Ping-Hsun Wu 

Academic Editor

PLOS ONE